# Initial Development of a Patient-Reported Experience Measure for Older Adults Attending the Emergency Department: Part II—Focus Groups with Professional Caregivers

**DOI:** 10.3390/healthcare11050714

**Published:** 2023-02-28

**Authors:** Blair Graham, Jason E. Smith, Pam Nelmes, Rosalyn Squire, Jos M. Latour

**Affiliations:** 1School of Nursing and Midwifery, Faculty of Health, University of Plymouth, Plymouth PL4 8AA, UK; 2Department of Emergency Medicine, University Hospitals Plymouth NHS Trust, Plymouth PL6 8DH, UK; 3School of Nursing, Midwifery and Paramedicine, Faculty of Health Sciences, Curtin University, Perth 6102, Australia

**Keywords:** emergency medicine, emergency department, aged, staff experience, focus groups

## Abstract

A wide range of healthcare professionals provide care for patients in the emergency department (ED). This study forms part of a wider exploration of the determinants of patient experience for older adults in the ED, to assist the development of a new patient-reported experience measure (PREM). Inter-professional focus groups aimed to build on findings from earlier interviews with patients conducted in the ED, by exploring professional perspectives on caring for older people in this setting. A total of thirty-seven clinicians, comprising nurses, physicians and support staff, participated in seven focus groups across three EDs in the United Kingdom (UK). The findings reinforced that meeting patients’ communication, care, waiting, physical, and environmental needs are all central to the delivery of an optimal experience. Meeting older patients’ basic needs, such as access to hydration and toileting, is a priority often shared by all ED team members, irrespective of their professional role or seniority. However, due to issues including ED crowding, a gap exists between the desirable and actual standards of care delivered to older adults. This may contrast with the experience of other vulnerable ED user groups such as children, where the provision of separate facilities and bespoke services is commonplace. Therefore, in addition to providing original insights into professional perspectives of delivering care to older adults in the ED, this study demonstrates that the delivery of suboptimal care to older adults may be a significant source of moral distress for ED staff. Findings from this study, earlier interviews, and the literature will be triangulated to formulate a comprehensive list of candidate items for inclusion in a newly developed PREM, for patients aged 65 years and older.

## 1. Introduction

Capturing older peoples’ experiences of emergency care is essential, not least since patient experience may be positively associated with improved clinical outcomes and greater patient safety [1]. Understanding vulnerabilities in patient experience may also be used to drive meaningful improvements in care [2]. The existing literature places emphasis on the importance of communication, managing wait times and providing a ‘frailty- friendly’ environment [3,4]. A meta-synthesis of patient experience in the emergency department (ED) proposes a needs-based framework for optimising patient experience, focusing on communication, care, waiting, physical, and environmental needs [5]. Interviews with twenty-four patients during an emergency care encounter confirmed the face validity of these themes amongst older people. Furthermore, interviews suggest the emergence of a new analytical theme consisting of ‘attitudes and values of the ED team’. (See Part I—Interviews with service users) Aside from building on the existing literature, these data will be specifically used to derive a list of candidate items for a patient-reported experience measure for older people attending the ED (PREM-ED). 

Older people are an increasing user group in emergency departments (EDs), comprising over a quarter of ED attendances in most developed countries. However, providing emergency care for older people represents a real challenge for healthcare professionals. For example, many major conditions encountered in the ED present subtly or atypically in older adults [6,7], and the high prevalence of multimorbidity, polypharmacy and pre-existing geriatric syndromes can make physical diagnosis challenging [8,9,10]. In addition, the frequent presence of frailty [11], cognitive impairment [12] and communication/sensory disturbances [13] demands that older people receive holistic care. The provision of complex geriatric assessment within the ED is well evidenced to reduce rates of hospital admission, ED length-of-stay and mortality [14]. However, issues such as under-resourcing and crowding present barriers to the provision of holistic care. As a result, ED care is often focused on resolving single organ dysfunction and optimising patient flow [15]. Although training and competency-based education requirements in geriatric emergency medicine have been published, these are yet to be widely translated into physicians’ working practice [16]. The importance of involving a multidisciplinary team has been emphasised within prominent guidelines [17,18], and although not yet widespread, discrete, nurse-led, geriatric emergency departments have been suggested to further improve care [19]. 

Patient interviews provide a recognised means to develop patient-reported measures [20] as they provide in-depth insights related to health care and beliefs [21]. Where acutely unwell and/or very frail patients are unable to participate in qualitative interviews, inviting emergency care professionals to share their perspectives assures this patient group is represented in the development of the PREM-ED. In addition, professionals may highlight important, additional determinants of patient experience not recalled or emphasised by older adults during interviews. In the context of developing the PREM-ED, professional focus groups capitalise on inter-disciplinary interactions to formulate new ideas about the determinants of experience for older people in the ED. Therefore, this study aims to explore emergency care professionals’ perceptions of delivering ED care to older people and, specifically, to determine whether any additional analytical themes emerge in the conceptual needs-based framework.

## 2. Materials and Methods

This study used inter-professional focus groups of healthcare professionals working in the ED. We used the COnsolidated criteria for REporting Qualitative research (COREQ) checklist to report the study [22]. Ethical approval was granted by the UK Health Research Authority (18/LO/1194) and institutional approval was granted from the University of Plymouth (17/18-973).

### 2.1. Research Team and Reflexivity 

The focus groups were led by a male-identifying academic emergency physician (BG) who had earlier conducted interviews with older adults attending the ED. As reported in part 1 of the study, data analysis was supported by a male-identifying professor in emergency medicine (JES) and a male-identifying clinical nursing professor (JML). Identical standards for rigour and trustworthiness were followed, including the use of reflexive notes and a consideration of the researchers’ insider perspectives as clinicians. The researchers were known professionally by some of the participants. However, assurance was provided from the outset that anonymity and impartiality would be respected. Participant information highlighted the purpose of the study for the development of the PREM-ED. 

### 2.2. Theoretical Framework

The methodological orientation of the focus groups was informed by a recognised definition of high-quality care [23], to investigate health professionals’ perspectives on the clinical outcomes, provision of safe care and desired patient experience. 

### 2.3. Study Setting

In order to maximise the potential for representative responses, focus groups were conducted at the ED of three hospitals in the South West of England. We selected one large teaching hospital which is also the regional major trauma centre, and two medium-sized district general hospitals with a mixed urban/rural catchment area. One of these hospitals had recently developed a specialist Older Peoples’ Emergency Liaison (OPEL) service, staffed by specialist nurses within the ED (Table 1).

### 2.4. Study Participants

A purposive sampling strategy was used to recruit participants to ensure an adequate representation from different professional groups and levels of seniority. An open invitation was issued by email to all clinical staff working in the EDs of the three participating study sites. Those interested were provided with a detailed information leaflet and consent form. Medical, nursing, allied health professional and ancillary staff who worked within the ED for at least six months were eligible for inclusion. Staff for whom the ED was not their permanent place of work, who had less than six months’ experience or who did not work in a patient-facing role were excluded. 

### 2.5. Data Collection

The focus groups (n = 7) were facilitated by the lead researcher (BG), who had previously received training in qualitative research methods. Participants were asked to self-report their age, staff group, level of experience and professional education level. The focus group schedule directly reflected the questions presented to patients during the patient interview study (Part 1), whilst also considering the need to obtain professional perspectives. Therefore, the use of a ‘question route’ [24] was proposed. This technique was used to facilitate the flow of constructive and in-depth conversations using principles of active listening [25]. Focus groups were audio recorded and field notes taken to capture non-verbal aspects of communication. Focus groups were conducted in a private room away from the operational ED setting. 

### 2.6. Data Analysis

Data analysis was conducted separately to the patient interviews, although followed an identical strategy as detailed in Part I of this study [26]. As such, the qualitative data analysis used framework analysis and a conceptual framework of patient experience developed by the authors was used. [5] Units of analysis were identified from written transcripts, using NVivo Version 12 (QSR International, Massachusetts, 2012). Data were then organised under the pre-existing analytical themes of the framework. 

## 3. Findings

### 3.1. Description of Participants

Thirty-seven participants were recruited, consisting of 20 emergency physicians, nine emergency nurses, three OPEL specialist nurse practitioners, two healthcare assistants, one physiotherapist, one occupational therapist and advanced clinical practitioner. Participants were pre-assigned to a focus group held within their locality. 

Focus group participants were more likely to be female (26/37; 70.2%). The length of the discussion ranged from 54 to 94 min per focus group, with an average length of 72 min. 

A summary of focus group characteristics is presented in Table 2.

### 3.2. Coding and Emerging Themes

Using framework analysis, a total of 150 unique statements were linked to existing analytical themes and sub-themes based on the conceptual framework (Table 3). Following its identification within interview data (Part 1), the new analytical theme—‘attitudes and values of the team’—was included within this analysis, and a further 20 linked statements were identified. 

### 3.3. Presentation of Findings

#### 3.3.1. Communication Needs

Healthcare professionals in all focus groups assigned a great deal of importance to giving adequate information to older adults. However, staff expressed frustration that time constraints and working pressures could undermine the desire to ensure that adequate information was provided. This could lead to a poor experience for patients.

*When we are under pressure we don’t have or allow enough time to explain the meaning of the attendance, and yes, we’ve focused on the diagnosis and ruling out conditions, which is good- but we might not have addressed their issue at all*.(Nurse, Site 02)

Healthcare professionals were also cognisant of patients’ ability to receive information whilst in the ED, due to issues such as sleep deprivation and anxiety amongst older adults. The provision of easily accessible written information was viewed as important, although staff understood the limitations for those with impaired eyesight or limited literacy. For this reason, an appropriate verbal introduction and description of role was viewed as essential.


*There are some aspects like ‘who’s who?’, that’s important, isn’t it? We’ve made a poster which should be in every cubicle showing team colours … you could make a very good argument that if a patient doesn’t have glasses … or even if they do … can they read it? So we’ve got to emphasise how you introduce yourself. Who you are, what you are …*
(Physician, Site 02)

Additionally, staff discussed the prevalence of hearing loss amongst older adults. This was something commonly encountered and could have a detrimental effect on the quality of communication. There was a general recognition that the ED represented an unfamiliar environment for most older adults, and that commonly used terminology could be misunderstood. The provision of adequate explanation was suggested by participants to overcome the confusion resulting from an unfamiliar environment and improve experience as a result.

*For us, ED is familiar … but to patients … they don’t realise that majors is majors and minors is minors. They’ve got nothing to help them understand. [I think] that’s a piece of work that needs to be done … [always] explaining where you are, what is going to happen, who is going to come and have some expectations of what is going to happen*.(Physician, Site 01)

Staff discussed the process and challenges of obtaining an accurate medical history. In particular, the repetition of questions and the use of jargon were viewed by participants as negatively affecting patient experience.

*Repetition of questions can be a problem. It’s not done intentionally but the level of communication between teams sometimes isn’t there and patients sometimes get upset that they’re being asked the same questions*.(Physician, Site 03)

When considering standards of interpersonal communication, the need to provide introductions was recognised as important in all focus group discussions. Positive communication with older adults could have an especially positive effect on those who were normally socially isolated. 

#### 3.3.2. Emotional Needs

The theme Emotional Needs identified ‘acknowledging uncertainty’, ‘recognition of suffering’, ‘empowerment’ and ‘reassurance’ as sub-themes.

Staff recognised that uncertainty was likely to characterise attendance for older people and could result from multiple aspects of the ED stay, including discrete aspects such as the decision to admit to hospital, arrangements being made at home—such as for pets—and implications for family life.

*The uncertainty of whether they’re going to be admitted. It’s a big deal for everybody, but especially the elderly patients who may have other considerations like frail elderly partners, pets, those sorts of things … complications with families … I think one of the really big anxieties is ‘am I going to be admitted’ ‘how long am I going to be in for’, ‘what are the knock-on effects for my family’*.(Nurse, Site 02)

Staff were adept at recognising suffering in older people across multiple dimensions, including fear surrounding possible death, anxiety around incontinence and access to toilet facilities.

*I have people trying to pull my uniform and say “I really need the toilet” because they haven’t been given a call bell, which happens very often, or because they’re in the corridor unaccompanied and all they can do is wave frantically for help to go to the toilet*.(Occupational Therapist, Site 01)

Staff viewed the empowerment of older adults as extremely important and an essential function of their role, but they discussed that many of the processes of emergency care could undermine efforts to involve older people as active participants in their care, and lead to disempowerment. This was recognised as something not only detrimental to the patient experience, but also leading to poorer outcomes of care.

*The minute you disempower somebody … you put them in an ambulance, and you ask them to wait for a period of time, you immediately disempower them so that they don’t care for themselves for that period of time and it doesn’t take long for that to become a longer lasting state and because we are in this environment where there isn’t enough space, it compounds the issue*.(Physician, Site 01)

Despite limitations imposed by the emergency care system, there was a clear desire to engage older people in decision making wherever possible. Staff discussed that doing so could have positive implications, especially with regard to discharge processes and longer-term care planning. Other clinicians recognised that agreeing solutions with the multidisciplinary team could help prevent deconditioning and harm arising from preventable hospital admission, even if this meant the acceptance of a degree of risk. For example, one participant recalled a case where an older male patient was empowered to exercise personal autonomy and make an informed discharge decision, even though the consequences of this could lead to death. Providing early decision making around whether to resuscitate patients at risk of deterioration is recognised as good practice. The importance of this as a critical decision for older people was discussed recurrently within many focus groups. Staff thought it was important to have informed and honest conversations with older adults, specifically regarding the limitations of care they could provide.

*Sometimes we undertake distressing things to elderly and frail people and the recognition of dying and [not providing] CPR as a normal course of death, and changing that so that people feel empowered to say ‘let’s sign this form’ … I know it doesn’t mean I’m not going to be treated but they know my ceiling of care’*.(Nurse, Site 01)

Staff recognised the potential role of accompanying persons, such as relatives and cares, in achieving patients’ needs. The presence of accompanying persons, such as relatives, were generally viewed as positive for older people when attending the ED. Participants acknowledged that relatives could provide advocacy and collateral history, as well as help guide treatment escalation decisions. Conversely, the presence of relatives could sometimes introduce complexity or, at worst, interfere with processes of care.

*I had a patient come into [the Resuscitation area] and the doctor came in to discuss the treatment escalation form and straight away the daughter said, ‘you don’t need to ask him anything, you can speak to me’ and she said straight away he’s for full resuscitation and the patient didn’t even get a look in. The doctor was still trying to talk to him [the patient], and she was butting in all the time*.(Healthcare Assistant, Site 03)

#### 3.3.3. Care Needs

From the healthcare professionals’ perspective, meeting care needs centred around patients’ individual expectations of care. Clinicians were cognisant of unique care needs resulting from physiological and anatomical changes because of ageing, for example when identifying occult injures in patients presenting with falls.

*There is a lack of recognition of the occult injury or reason for presentation underlying injury in these patients … there’s lots of evidence out there to suggest we are not assessing the underlying reasons that have brought elderly patients to us … comorbidities, polypharmacy, the home situation, that sort of thing*.(Advanced Clinical Practitioner, Site 01)

Participants discussed that their own professional guidelines did not always account for the needs of older adults. Specifically, process/time bound targets were often viewed as inappropriate and a potential barrier to the provision of holistic care.

*If you look at things like the sepsis guidelines, they are very focused on fast tracking children but not the elderly … [the elderly] just get lumped with adults. But actually a 20-year-old is very different from a 90-year-old*.(Physician, Site 03)

There was also a recognition that, on some occasions, the correct approach could be to do nothing, withdraw care, or facilitate end of life care within the ED. As opposed to being futile, such encounters were viewed as positive by clinicians, allowing them the opportunity to facilitate person- and family-centred care.

*So we had a patient last week … and we stopped [active treatment] … well, we spoke to the patient and asked her what she’d like and she said ‘I want to go to sleep’ so we tucked her up in bed. She’s expressed her wishes and then we planned to discuss with the family about her expectations and agree a plan. And we signed the treatment escalation form and admitted her for end-of-life care*.(Physician, Site 03)

Participants recognised the overall value of capturing and measuring outcomes that are meaningful to patients, for example through the collection of patient-reported experience and outcomes measures.

*I think achieving good clinical outcomes goes back to what the patient actually wants. For patients, going back to the PREMs [Patient Reported Experience Measures] is about looking at the clinical outcomes they actually want. So, if the pain gets under control so they are able to mobilise then that should be a good outcome*.(Advanced Clinical Practitioner, Site 03)

#### 3.3.4. Physical and Environmental Needs

Ensuring that patients’ basic physical needs were met, facilitated by a welcoming physical environment, was viewed as paramount to ensuring quality care by participants. 

*Most older people are not interested in their physiology, they are not particularly interested in having a lactate taken within seconds of arrival. They are interested in whether a window is there … whether there is a clock … and if the nurse offered them a cup of tea* (field note: agreement from group).(Physician, Site 03)

There was general recognition that essential needs included fluids, food and toileting. However, staff recognised that these basic needs were often not met, due to a lack of resources, constraints on staff availability and the physical layout of the ED itself.

*There’s a general lack of dignity … a lack of privacy. It’s not a ward here, but it gets used like a ward because people are here for hours and hours … it’s not satisfactory, is it? We’d like to give everyone a sandwich, but often we run out of them. We run out of chairs these days*.(Healthcare Assistant, Site 02)

Staff were also able to give examples where a lack of basic needs provision, such as hydration, may have contributed to clinical outcomes and need for admission.

*Dehydration certainly makes it harder to discharge patients. Concentration, memory and focus all decrease*.(Occupational Therapist, Site 01)

Participants agreed that the physical environment often presented a barrier to facilitating a positive patient experience. In particular, the ambient environment could present patients with a range of noxious stimuli, including excessive noise, bright fluorescent lighting and even exposure to interpersonal violence and aggression. Clinicians also recognised that, in addition to providing a poor experience, such an environment could have a negative effect on clinical outcomes, by increasing the incidence of disorientation and conditions such as delirium.

*I think the [ED] environment must be very distressing for them [older people] … the hustle, bustle, police being around, monitors, alarms and swearing*.(Nurse, Site 03)

Use of monitoring equipment, manual handling equipment and furniture not designed specifically for older people—such as ED trolley gurneys—were also viewed as both a barrier to patient experience and a risk to patient safety.

#### 3.3.5. Waiting Needs

Clinicians shared a common concern that waiting could be a frustrating experience, and the need to provide accurate information on waiting times was commonly recognised. Although anecdotal, a consensus emerged that older adults were more likely to tolerate protracted waiting periods without expressing dissatisfaction, compared to some other groups. However, this tolerance was perceived as potentially problematic. 

*The elderly patients are least likely to make a fuss … so they’re most likely to be forgotten. Whilst you are busy with somebody who is exhibiting challenging behaviour and running amok in the department, the poor [elderly patient] in the corner has wet herself because she can’t reach her call-bell, or she lacks capacity to make herself heard*.(Nurse Practitioner, Site 02)

There was an appreciation that, for some older adults, the uncertainty around waiting could be distressing especially when the surrounding environment was busy or fractious. Strategies for overcoming the boredom, frustration and inefficiency arising from waiting included methods to provide distraction. Such interventions could be simple, such as the provision of reading materials or jigsaws, or more advanced, comprising the use of technology. Waiting to front load other forms of assessment, alongside a medical ‘workup’, was seen as a better way to utilise time. 

*We could issue books to read, newspapers, a jigsaw puzzle. There are so many things you could think of introducing … to improve the experience*.(Physician, Site 03)

There was a widespread recognition of problems for older adults associated with crowding. Corridor care was viewed as an inevitable, albeit unacceptable, feature of the emergency care journey, which was an affront to patient dignity and could threaten patient safety. Clinicians expressed empathy with patients who had to endure corridor care, and they felt disempowered to change the situation. Some participants expressed that a sense of failure—or even shame— was caused by working in crowded environments 

*Sometimes it’s like being on double beds, isn’t it? The patients are coughing all over each other, so you fail on every nursing element ever imaginable … you fail on dignity, on infection control … every single thing that you’ve ever learnt you’ve failed on because of the crowding. And there’s not an awful lot you can do about it*.(Advanced Clinical Practitioner, Site 01)

#### 3.3.6. Attitudes and Values of the Team

Focus groups participants frequently discussed their role in the wider ED team. There was a recognition that, in the ED, nursing staff were often conflicted between providing basic patient care, technical tasks such as venepuncture and cannulation, and facilitating patient flow. This often meant that no one was allocated to meeting patients’ essential needs, despite the recognised vital importance of this function. As a solution to this problem, there was an agreed expectation amongst participants that all members of the ED team responded to meeting patient’s essential care needs. The sense of a team ethos and flattened hierarchy that resulted from this was generally viewed as desirable and helped ED staff to develop a shared team ethos.

*Patients have often said to me ‘look at that doctor making the bed!’, and I’m like ‘yeah, that’s called teamwork, that’s what we do’. And patients and relatives are often surprised that doctors do basic care as well*.(Nurse, Site 02)

Conversely however, the onus on senior clinicians to perform basic care tasks could be a distraction from their core role, affecting professional efficacy.

*In the past four months I have done more that I would deem ‘out of my doctor role’ because the nurses are short staffed. I am regularly giving medications and getting commodes and urine dips and urine bottles and fluids and stuff. Which is fine … but that then actually impacts on our role as doctors and what we can achieve. And we don’t have the time either so what you’ve got is a group of professionals who are each interplaying with each other’s job roles, and no-one is taking responsibility for that person’s care*.(Physician, Site 01)

Specialist team members who could provide care for older adults were praised by many participants. One department had established an ‘Older Peoples’ Assessment and Liaison’ Service consisting of senior nurse clinicians. This service provided the holistic assessment of older people in the ED and was held up as a model of exemplary practice by participants. Input from the wider multidisciplinary team, including Physiotherapy and Occupational Therapy, was also viewed as beneficial in assisting clinical decision making for older adults and increasing patient satisfaction.

*If you go back a few years, we didn’t have therapy/OT [Occupational Therapy]/MSK [Musculoskeletal] practitioners at the front door. We now have a true MDT at the front door, and we know that’s the right thing for older people. And everyone’s happy to make their own decisions, and quite often if the therapy team is happy with their mobility, then they go home. And I think the patients like that*.(Frailty Nurse Specialist, Site 03)

### 3.4. Supplementary Themes

Three supplementary themes arose, separate from the framework analysis. These are ‘staff distress’, ‘recognising older people as a vulnerable user group’ and ‘views on provision of geriatric ED services’. Whilst not directly linked to the original aims of the study, staff were keen to discuss their lived experiences of caring for older people in the ED, and the effect on their professional values and identity. These themes, therefore, provide an important contextual insight into the challenges faced by healthcare professionals when providing care to patients in the ED.

#### 3.4.1. Staff Distress 

Although not the original focus of this investigation, focus group participants reported significant distress when unable to provide older patients with a desirable experience.

*I just feel really guilty: sometimes you have a choice between providing medical care or providing compassionate care*.(Physician, Site 01)

During the focus groups, it was common for health professionals to compare their occupational experience of providing care with standards they would desire for their own loved ones or themselves. Such descriptions revealed a deep empathy for older people and revealed feelings of distress when the desired standards were not met.

*Sometimes you just feel ashamed. The poor patient is on a commode in a cubicle … it’s just … you wouldn’t want to be in that situation … you wouldn’t want your mother to be in that situation*.(Physician, Site 01)

For some participants, the presence of service pressures and an emphasis on patient flow increased feelings of distress and could lead to conflict with others in the team.

*We’re the first point of contact for that patient coming in but we seem to be the last people to be drip fed any sort of budget … where we can make holistic improvements? You know, they’re talking about redesigning and remodeling and rebuilding but … it’s just simple things we need, like, basic human comforts*.(Healthcare Assistant, Site 02)

In contrast, one doctor was able to reflect positively on the effort that her department made to care for older adults.

#### 3.4.2. Recognising Older People as a Vulnerable User Group

Despite perceived shortcomings in delivering healthcare to older adults, professionals identified that acquiescence was commonplace, with older adults appearing to tolerate suboptimal care and undesirable clinical outcomes, when compared to other patient groups. This could make them more vulnerable to experiencing poor care.

*Yeah, I find that they don’t want to trouble you as much as other patient populations … so they are sitting in pain for longer perhaps, and they don’t want to ask to go to the toilet. I had one chap who even wet the bed, because it was so busy in the department he didn’t want to trouble anyone because he saw it as a minor problem*.(Physician, Site 01)

One nurse viewed it as her responsibility to recognise older peoples’ tendency towards acquiescence and ensure patients had an advocate.

*Older people don’t always have a voice, do they, not like the younger generation. They will just sit quietly and wait patiently … so it’s making sure there’s an advocate for them. They don’t always have family … someone to stand up for them*.(Nurse, Site 01)

#### 3.4.3. Views on Emergency Care Systems for Older People

As well as identifying problems and perceived deficiencies in ED care for older people, focus group participants discussed their ideas for improving service configuration. Views regarding the ideal configuration for geriatric emergency care were mixed. Some favoured highly specialised services and designated areas for older people. Conversely, others recognised that due to demographic shifts, older adults were likely to become the predominant user group of ED services, and that the configuration of the department should reflect this.

*Do you not make the older patients your core user group and others have to fit in around this, especially as we know this population is going to skyrocket in the future. I think it should be more focused on the elderly … you should make that your core business and figure out how others fit around it, rather than put them aside*.(Physician, Site 03)

Finally, one emergency physician reflected on visiting a specialist geriatric emergency department which had recently opened in another locality. Although not commonplace in UK practice, this was seen as a gold standard model of care.

*At [locality] they’ve just recently opened a geriatric ED run by a geriatrician. It runs from 8am to 10pm and everybody over 75 who goes into majors is seen in that area. They have a particularly high percentage of older patients. But they’ve really grabbed the bull by the horns by creating a separate environment and it’s a nice structure to ensure there is daylight and more privacy and people are more oriented. And I think that’s the way to go*.(Physician, Site 03)

## 4. Discussion

The findings of our study provide some insight into professionals’ experiences of caring for older adults in a UK ED setting. No new analytical themes or sub-themes resulted from framework analysis. As such, this study further validates the existing needs-based conceptual framework developed from the literature [5] and expanded using interviews with patients (Part I). However, staff offered additional perspectives in relation to the existing themes. Staff were attentive to the need to establish rapport and effective communication with older adults and prioritised identifying and meeting care needs that were tailored to the individual patient. This included not only the delivery of medical interventions such as investigative procedures and analgesia, but also making decisions on treatment escalation and providing effective supportive care including dignified end-of-life care, where this was deemed the most appropriate option. Providing palliative care to older people has previously been recognised as an emerging and important function of emergency clinicians, despite the presence of barriers such as sub-optimal access to information, collateral history and time constraints [27]. As well as recognising the need to provide high-quality medical care, staff were intrinsically motivated to meet the most essential needs of older patients, such as providing ready access to food, drinks and toileting. The importance of providing a comfortable environment and protecting older peoples’ dignity was explicitly recognised. However, participants reported that crowding and corridor care were commonly encountered and presented barriers to achieving the desired standards of care. Such concerns are likely to be well-founded, with the effect of ED crowding on mortality and patient experience being well documented in the literature [4,28]. 

Teamwork has been previously recognised as paramount to improving patient safety, reducing clinical errors and improving the efficiency of care in the ED [29]. Conversely, hierarchical power relationships—defined as relationships being “based on power from … expertise or experience”—may present a barrier to team co-operation, cognition, co-ordination and interdisciplinary exchange [30]. In our study, there was a notable sense of solidarity and team spirit amongst the different members of the ED team, who recognised their common purpose in helping to achieve an optimal experience for older people. Although focus groups included members of staff at varying levels of seniority, no evidence of hierarchical power relationships was displayed within the group interactions. 

This study builds on the findings of existing international research exploring nurses’ perspectives of caring for older adults in the ED. Findings from this study provide a meaningful addition to this literature by incorporating perspectives from other groups of staff members, such as emergency physicians and allied health professionals. Within existing studies, Kihlgren et al. [31] interviewed nurses in a Swedish ED and determined that a focus on providing medical procedures threatened the provision of holistic nursing care to patients. Such frustrations were reflected by clinicians from all professional backgrounds in our study. In addition, inter-professional discussions revealed a shared concern regarding the appropriateness of the ED as the optimal location for many older patients. Lennox and colleagues [32] undertook focus groups in an Australasian setting, which identified limitations in provider knowledge, the suitability of equipment and environment, and a limited time for discharge planning as barriers to providing high-quality care in the ED. Content analysis of over five-hundred survey comments from ED nurses was undertaken by Boltz et al. [33], revealing the perceived importance of establishing effective communication with older people, having time to perform care, and fostering a safe and enabling environment. These themes were reflected by participants in the focus groups, highlighting their relevance to all those delivering ED care to older people, irrespective of professional identity, in the UK setting. 

Staff were cognisant that experiences and subsequent clinical outcomes for older adults are likely to be related and gave examples where the two were linked—this included the provision of adequate drinks (hydration) in preventing delirium [34]. Even though recent efforts have been made to suggest standards of care for older people attending the ED [16], participants in multiple groups drew comparisons with standards achieved for children within their ED. This demonstrated a perceived sense of inequality for older adults, who were recognised by participants as a vulnerable and a discrete user group. Despite this, participants did not reach a clear consensus on the benefits of bespoke services delivered by sub-specialists in ‘geriatric’ EDs, over and above improving access to appropriate facilities and training within their general ED environment. The co-design of services with staff and service users has previously been recognised as beneficial for improving the design of emergency care environments, and this is likely to be an area that would benefit from further investigation, specifically for older adults [35]. However, even when perceived to be desirable, whole-service reconfiguration may fail to improve either access to care or clinical outcomes and give rise to unintended consequences [36]. Smaller-scale quality-improvement (QI) initiatives can be initiated at a service provider or departmental level and promote rapid systematic change following the identification of a problem [37]. The QI approach is commonly advocated within emergency medicine [38] and has been demonstrated to improve processes of care and experience for patients attending the ED [39,40]. Chartier et al. state that problems to be addressed by QI methods should be important, occur frequently, demonstrate deficiency, and be realistic to address [41]. To this end, problems identified by ED staff—such as improving access to fundamental care, toileting and environmental optimisation—should be considered a high priority. Additionally, the original needs-based conceptual framework includes a range of pragmatic suggestions for improving older peoples’ ED experience [5]. 

Despite participants’ best intentions, the focus group discussions revealed a significant gap between the standards of care desired for older patients and what was often achieved. Participants could recall instances of suboptimal care and reported feelings such as guilt and shame. Moral distress is a recognised phenomenon resulting from an inadequate working environment [42], and has been defined as “when one knows the right thing to do, but institutional constraints make it nearly impossible to pursue the right course of action” [43]. Following a meta-ethnography of the existing literature exploring emergency and critical care nurses’ experiences of moral distress, Arnold [44] conceptualises an ‘internal battle’ as a metaphor for “moral distress as the nurses described their internal conflicts of conscience with doing what they are told to do versus what they feel is the right thing to do”. Sub-themes including the presence of challenging environments; feelings of anger, despair and guilt when experiencing moral distress; and effects on personal and professional relationships, were reflected by participants in these focus groups. As moral distress can have sustained effects on staff wellbeing [45], the recruitment and retention of staff [46], and patient safety [47], mitigating its effects amongst those providing care for older patients in the ED is essential.

### 4.1. Relevance of Focus Group Findings to Development of PREM-ED 65

The patient interview study accompanying this paper has confirmed that older peoples’ experiences of ED care can be categorised according to a ‘needs-based’ conceptual framework, with the addition of a new theme of attitudes and values of the team. (Part I) Framework analysis of the focus group findings further confirms this concept, reflecting and reinforcing themes encountered in the existing literature and reported by patients. Moreover, focus groups provide valuable additional insights into the experiences of older adults, delivered through the critical lens of the healthcare professionals. Specifically, older adults were noted, in general, to report positive aspects of their experience during patient interviews. Conversely, the healthcare professionals in the focus groups were forthcoming in revealing perceived weaknesses and vulnerabilities in processes of care. Professionals’ candid insights into the challenges faced when caring for older people in the ED are useful in highlighting determinants of a suboptimal patient experience. Such vulnerabilities may be measurable and indicate important areas for item development within PREM-ED 65. In addition, an important function of the focus groups is to ensure that the views of groups of older adults potentially under-represented in the interviews, such as very frail patients and those requiring end-of-life care, are also considered when generating a list of candidate items.

### 4.2. Limitations 

Our study has several limitations to address. Despite attempting to capture a sample of healthcare professionals from across the multidisciplinary team, the views expressed in our focus groups may not be fully representative of all staff working in the ED. Another limitation is that we observed that participants were more likely to be female, and that physicians were overrepresented compared to other professions. Although we did not overtly observe dominant relationships during the focus groups, it is possible that this may have influenced expression of views by some other participants. Furthermore, our study was conducted in one geographical area of the UK and may not account for regional variations elsewhere or in other countries. Finally, another limitation relates to the nature of qualitative research methods, where the findings are not considered generalisable. Nonetheless, our study provides an addition to the existing body of literature in this area, offers some unique insights into the personal and professional challenges encountered by staff when caring for older people in the ED, and will help inform item generation for PREM-ED 65.

## 5. Conclusions

Interdisciplinary focus group discussions with ED staff further confirm the existing needs-based framework. This framework provides a basis for conceptualising the determinants of older peoples’ experiences of care in the ED. Irrespective of seniority or their professional role, staff prioritise the provision of appropriate communication for older people, whilst identifying and meeting individual care needs. Although not always possible, staff have a desire to meet patients’ basic human needs within an appropriate physical environment.

In addition, our study has also highlighted important supplementary themes. Specifically, a gap frequently exists between the desired standards of care and those delivered in the real-world setting. Importantly, the findings demonstrate that this may result in significant moral distress for providers. By capturing patients’ real-world experiences of ED care, PREM ED 65+ may provide a powerful means of identifying such vulnerabilities where they exist, so that improvement can result and these effects are mitigated.

## Figures and Tables

**Table 1 healthcare-11-00714-t001:** Study settings.

Site Identifier	Site Characteristics
**Hospital 1**	Type 1 ED Major Trauma Centre 100,000 attendances per annum
**Hospital 2**	Type 1 ED Regional Major Trauma Unit 80,000 attendances per annum
**Hospital 3**	Type 1 ED Regional Major Trauma Unit Specialist OPEL service 80,000 attendances per annum

ED = Emergency Department; OPEL = Older Peoples’ Emergency Liaison Service

**Table 2 healthcare-11-00714-t002:** Focus group characteristics.

Hospital Number	Focus Group Number	Duration hh:mm	Total n Participants Occupational Group, n	Gender, n
01	1	01:18	Total 6 Physician, 5 Nurse, 1	Male, 2 Female, 4
01	2	00:54	Total 5 Physician, 4 Nurse, 1	Female, 5
01	3	01:09	Total 6 Physician, 3 Nurse, 1 Therapist, 2	Male, 1 Female, 5
02	4	01:34	Total 4 Physician, 1 Nurse, 1 ACP, 1 HCA, 1	Male, 2 Female, 2
02	5	01:24	Total 5 Physician, 2 OPEL Nurse, 3	Male, 1 Female, 4
03	6	01:00	Total 6 Physician, 2 Nurse, 3 HCA, 1	Male, 3 Female, 3
03	7	01:06	Total 5 Physician, 3 Nurse, 2	Male, 3 Female, 2

OT = Occupational therapist; PT = physiotherapist; HCA = healthcare assistant; ACP = advanced clinical practitioner; OPEL NP = older peoples’ nurse practitioner.

**Table 3 healthcare-11-00714-t003:** Analytical themes and sub-themes.

Analytical Theme	Existing Sub-Theme	New Sub-Theme	Supplementary Theme ^1^
**Communication Needs**	Interpersonal Communication Informational Communication	Social Communication	
**Emotional Needs**	Acknowledging Uncertainty Recognising Suffering Providing Empowerment	Reassurance	
**Care Needs**	Symptom Relief Procedural Care	Responsiveness	
**Waiting Needs**	Impact of Crowding	Waiting experience	
**Physical/Environmental Needs**		Fundamental Needs Equipment and Devices	
**Attitudes and Values of the team (new)**	-	Perceptions of teamwork Staff attitudes and professionalism	
**Supplementary Theme ^1^**			Staff distress Recognising older people as a vulnerable user group Views on emergency care systems for older people

^1^ Supplementary themes arose in addition to framework analysis and are related to participants’ lived experience of caring for older adults.

## Data Availability

The data presented in this study are available on request from the corresponding author.

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
