# Peer review of "Initial Development of a Patient-Reported Experience Measure for Older Adults Attending the Emergency Department: Part II—Focus Groups with Professional Caregivers"

_healthcare, 2023, doi:10.3390/healthcare11050714_

Round 1

Reviewer 1 Report

I don't understand if in your study (that explore the professional perspectives on caring for older people) did you find a different result than patient reported experience?

Reviewer 2 Report

Dear authors,

Thank you very much for this very interesting draft. The qualitative study aims to develop a patient reported experience measure (PREM) for older adults in the ED in the UK. This is a highly important task thinking of older people in the ED waiting for hours and in risk of dehydration, delirium and falls, e.g. 

The researchers performed 7 focus grups with 37 ED professionals in three hospitals in South West England. The focus groups consisted mainly of physicians (20 of 37 focus group members).

The interesting results are presented very well, I have read the article with tension. 

One limitation is obvious to me and should be mentioned and thus a correction be made. The physicians were the majority in the focus groups, the discussion (page 12, line 476-480) focusses on research with nurses in ED. Please mention and put in context with the discusssion. 

Reviewer 3 Report

This study investigated the opinions of health providers in the emergency department on delivering care to older people and explored whether any additional analytical themes emerge in the conceptual needs-based framework. Research about providing optimal care for older patients in the emergency department is needed to guide current clinical practice. I have some comments for the authors.

1. It would be nicer if the authors summarized their findings in a table, if possible.

2. Please consider discussing more potential methods to improve suboptimal care for older people in the emergency department.

3. Please define the abbreviations in the manuscripts upon first use.

Reviewer 4 Report

very interesting qualitative type of study. i wish the sample size was more. Definitely, more quality improvements would benefit the older adults especially in the emergency room settings. 

Some suggested changes:

Line 87 a male researcher , Why Define the gender of the researcher ? May be a subjective or personal bias like gender Bias? I would suggest to remove "Male" and add another adjective like a clinical or adult or certified. 

Lines 160, 168 older people should be replaced with Older patients or Older adults because we are talking in scientific terms 

Round 2

Reviewer 1 Report

thanks for your answer

Author Response

Thank you very much for your response. 

With Kind Regards, 

Dr Blair Graham, Lead Author